# The BABITT questionnaire for evaluation of bowel and bladder function in children who are introduced to assisted infant toilet training - content validity and feasibility

Anna Leijon[1,2,3*], Terese Nilsson[1,2,3], Ulla Sillén[4], Anna-Lena Hellström[5], Linda Vixner[6ᴓ], Barbro H. Skogman[2,3ᴓ]

1 Department of Family Medicine, Region Dalarna, Falun, Sweden, 2 Center for Clinical Research Dalarna - Uppsala University, Falun, Sweden, 3 Faculty of Medical Sciences, Örebro University, Örebro, Sweden, 4 Institute of Clinical Sciences, Sahlgrenska Academy, University of Gothenburg, Gothenburg, Sweden, 5 Institute of Health and Care Sciences, University of Gothenburg, Gothenburg, Sweden, 6 School of Health and Welfare, Dalarna University, Falun, Sweden

ᴓ These authors contributed equally to this work.
* anna.leijon@regiondalarna.se

## Abstract

Functional bowel and bladder disorders are prevalent among children. In 2019 our research group launched the BABITT study (Bowel and Bladder function in Infant Toilet Training), a randomized intervention study to investigate whether introduction to assisted infant toilet training reduces the prevalence of functional bowel and bladder disorders in children up to 4 years of age. Diagnostic criteria for gastrointestinal disorders are defined by the ROME Foundation, while the International Children's Continence Society (ICCS) provides definitions of functional bladder disorders. Preceding the larger ongoing BABITT study, the aim of this present observational study is to construct, assess content validity and evaluate feasibility of a questionnaire for parent report.

### Methods

A web-based questionnaire was developed in three consecutive steps. In Step 1, the questionnaire was outlined based on literature review and expert panel discussions. In Step 2, the questionnaire was validated for relevance and simplicity by content validity index (CVI) using 4-point Likert scales. With dichotomized data, an index level ≥ 0.78 was considered as acceptable. In Step 3, the respondent burden was analysed and a pilot phase allowed for evaluation of feasibility in the clinical study setting.

### Results

In Step 1, the Rome IV criteria and ICCS frameworks were selected for items comprising the primary outcomes in the BABITT study. After the final assessment round in

**Data availability statement:** Data for this study contain potentially identifying or sensitive personal information. Public sharing of this data may violate participant confidentiality in conflict with Swedish Research Review Authority approval. Data will, upon reasonable request, be de-identified and made available from Region Dalarna for purposes such as education, research and innovation. Requests will not be considered until after the last publication based on the data set has been finalized. Requests will only be considered if they are in accordance with all applicable data protection and privacy regulation according to the European General Data Protection Regulation Services (GDPR). Contact for request Region Dalarna, Sweden: forsknings.utlamnande@regiondalarna.se.

**Funding:** The work of conducting the study was funded by grants from the Centre for Clinical Research Dalarna, Region Dalarna (CKFUU- 961469, CKFUU-933634, CKFUU-967926) and the Regional Research Council of Uppsala – Örebro (RFR-967829), as well as the SwedishEnuresis Academy, The Samariten Foundation for Paediatric Research and The Mayflower Association. Sponsors and funders have not been involved in the design or conduct of the study nor in analysis of results

**Competing interests:** The authors have declared that no competing interests exists.

Step 2, the item-level content validity index (I-CVI) was excellent, ranging from 0.88 to 1.00 in most items, in all domains, for both relevance and simplicity. In the pilot phase Step 3, the response rate was 95% and the parents' acceptance of replying to the questionnaire was satisfactory.

## Conclusion

A web-based questionnaire was developed to evaluate parent-reported bladder and bowel function in children who are introduced to assisted infant toilet training. The BABITT questionnaire emerged as valid and feasible in its context.

## Introduction

Functional disorders and symptoms from the bowel and bladder are common in the pediatric population [1–4]. Despite their benign character, they are known to cause stress and suffering in the affected children and families[5,6]. However, it is crucial to acknowledge that self-reported bowel and bladder symptoms can be inconsistent and unreliable as recall bias and anxiety about the problem can result in the downplaying or catastrophizing of symptoms. Consequently, using robust tools is fundamental to allow researchers to quantify baseline variables, measure the effects of an intervention and track changes over time [6]. Awareness that the feasibility of instruments, questionnaires and interventions may alter when applied to complex study settings is essential, as is the importance of culturally sensitive translations to maintaining the validity of a tool [6,7].

In recent decades, the age at which toilet training typically begins has increased in Sweden, as in most industrialized countries [8–10]. It has been suggested that this may contribute to an increasing prevalence of bladder and bowel disorders [11,12]. At the same time, cases of stool toileting refusal, a dysfunctional elimination behavior closely associated with constipation, have been described. It presents in toilet-trained children who ask for a diaper before they release stools, despite using the toilet to urinate [13,14]. Traditional techniques of assisted infant toilet training, where the parent responds to elimination signals by helping the baby relieve itself in a potty or basin instead of a diaper, have been shown to enhance development of bladder control [15,16].

Functional gastrointestinal disorders encompasses a variety of conditions such as infant colic, infant dyschezia and functional constipation, presenting with recurrent symptoms unexplained by structural or biochemical abnormalities in the gastrointestinal tract. The ROME foundation, holding a paramount position in the field of gastrointestinal research, offers continuously updated symptom-based diagnostic criteria for functional gastrointestinal disorders, and is used a as gold standard in research. The present, fourth version (Rome IV) provides questionnaires for infants and toddlers [17].

Lower urinary tract dysfunction is a broad term with subsets of storage and voiding symptoms from the bladder. The International Children's Continence Society

(ICCS) is an organization aiming to establish standardized terminology and provide definitions of symptoms as well as diagnoses and sub-diagnoses of bladder dysfunction [2,17]. Validated questionnaires to assess bladder dysfunction are available. However, although some questionnaires include children from the age of 3 years, there are none for children from the age of toilet training to 4 years of age. The ICCS does not produce questionnaires; but rather offers guidance in selecting from the appropriate instruments [6]. In the field of urinary tract research, it is well known that the intricate function of the bladder may be affected by bowel dysfunction. It is suggested that mechanical impact in the small pelvis as well as the shared embryonic origin with partly common innervation account for this influence, though the pathophysiology is not entirely known [18,19]. The term BBD (bladder and bowel dysfunction) is commonly used to describe this parallel dysfunction [17], and the intimate connection between BBD and recurrent urinary tract infection has been increasingly highlighted [2,20–22].

The impact of toilet training on bowel and bladder function remains a hot topic for discussion. In 2019, the BABITT (Bowel and Bladder function in Infant Toilet Training) study, a randomized, parallel, open-label, two-armed intervention study was launched aiming to evaluate whether the prevalence of functional gastrointestinal and urinary tract disorders can be reduced by parents learning and practicing assisted infant toilet training with their baby [23]. In the ongoing BABITT study participants were randomly allocated to groups which initiated toilet training either at 0–2 months of age (intervention group) or at 9–11 months of age (controls). Bowel and bladder function up to 4 years of age is to be evaluated and the parental experiences of assisted infant toilet training will be described.

As there was no existing coherent instrument matching the complex scope and setting of the BABITT study, a web-based questionnaire containing Rome criteria and ICCS definitions for a parent proxy report in Swedish needed to be constructed. The aim of this present observational study was to construct and assess content validity and feasibility of the BABITT questionnaire, an instrument for evaluating bowel and bladder function in children who are introduced to assisted infant toilet training.

## Methods

### Study design

An observational study design with expert assessments was used to evaluate the content validity and feasibility in this present study. The questionnaire was developed in three consecutive steps, illustrated in Fig 1. In Step 1, the main elements in the field of interest were identified by literature review and expert panel discussions. In this way, the item pool was created and the draft questionnaire outlined in an iterative draft-review process. In Step 2, the questionnaire was validated in two assessment rounds by the CVI method and subsequently revised after each loop [24,25]. The BABITT questionnaire was finalized in a last round of discussions, refinement and approval, by the expert panel. In Step 3, after implementation online, a pilot phase assessed the feasibility of the final questionnaire. A detailed description of the larger, ongoing, randomized BABITT study is provided in the protocol article [23] and the full clinical study protocol is available at ClinicalTrials.gov (NCT04 082689).

### Expert panel, raters and pilot phase participants

The expert panel in Step 1 consisted of five of the authors of this paper. Two members were professors emeritae (ALH, US) with scientific and clinical expertise in the field of toilet training and bladder and bowel dysfunction, the former (ALH) with a background in urotherapy and the latter (US) in pediatric urology. One member was associate professor (BHS) with theoretical and clinical specialization in pediatric bowel and bladder disorders, and the last two members were PhD students (AL, TN) specializing in family medicine. The expertise of the senior members (AHL, US) and the recurrent discussions among all members of the expert panel were crucial cornerstones throughout the development process.

In Step 2, a group of independent raters (n=9) volunteered to assess the content validity of the proposed questionnaire. Raters were strategically recruited from hospital staff and personal contacts, and possessed wide-ranging professional

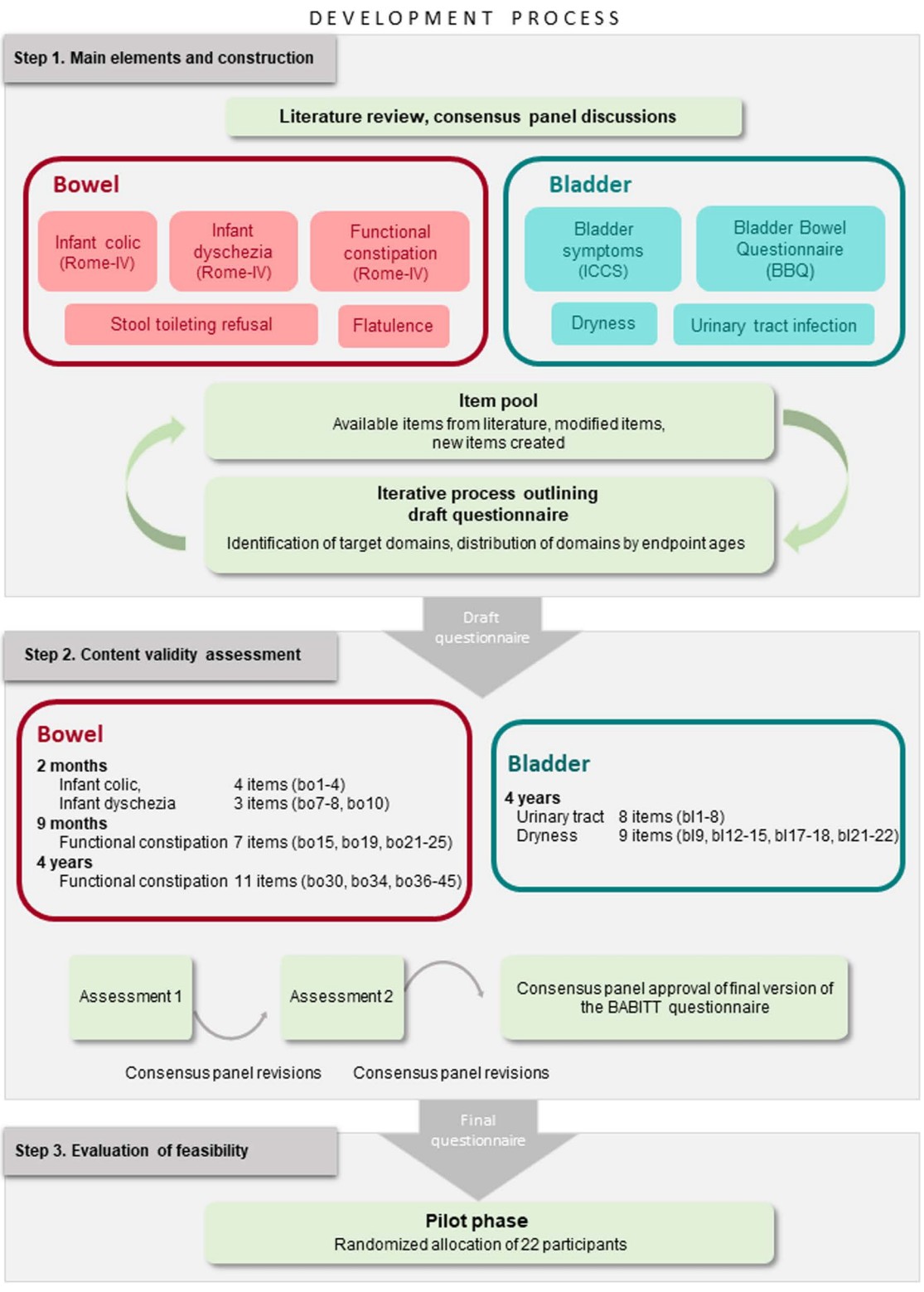

**Fig 1. The development process of the BABITT questionnaire.** The three steps of the development process are graphically visualized. The construction process and main elements in Step 1. The CVI process in two assessment rounds and a final revision in Step 2. The evaluation of feasibility of the final questionnaire in the pilot phase in Step 3.

experience, relevant to the study, as shown in Table 1. One eligibility criterion was that the rater should represent the target population - parents with personal experience of toilet training. Four raters had substantial experience of assisted infant toilet training [24]. One rater was excluded after the first assessment round due to a lack of nuanced input (all items were uniformly rated, without further explication). Accordingly, the remaining group was reduced to eight raters (n=8) [25].

In Step 3, full-term, healthy infants and their parents were eligible to participate in the pilot phase and 22 infants were enrolled. One parent per family answered the questionnaire on their personal computer, when the children were aged 2, 3, 6, 9 and 12 months [23].

## Step 1– Main elements and construction

Firstly, a small-scale qualitative interview study of parents and nurses at Child Health Centres was conducted by two of the researchers (AL, TN) [26,27]. This was in order to explore perceptions of the toilet training process in general and knowledge of assisted infant toilet training in particular. Secondly, the database PubMed was searched for articles using the following relevant MeSH terms: "constipation", "lower urinary tract symptoms", "toilet training" and "colic" as well as the key words "infant dyschezia" and "bladder and bowel dysfunction". The initial search string yielded 116 articles, which were aggregated and their abstracts read, to facilitate further selection. No systematic review was conducted, due to the timeframe. Any discrepancies were resolved through expert panel discussions. Relevant articles were chosen and included in Step 1. Out of this material, *Bowel*, *Bladder* and *Assisted infant toilet training* were identified as three main elements to constitute the questionnaire (Fig 1). Additional items on adherence to intervention and relevant background information on, for example feeding, parental education and family situation were added to the questionnaire and the content validity assessment, although they were not reported in this present article. Two of the researchers (AL, TN) created the initial item pool and the questionnaire was designed in an analysis-driven, iterative draft-review process and through expert panel discussions. The items were distributed in three domains with the same denominations as the main elements and outlined to match the BABITT study at ages 2 and 9 months as well as 2.5 and 4 years [23,25]. The expert panel approved the draft version of the BABITT questionnaire.

## Step 2– Content validity assessment

The raters completed two assessment rounds between October 2018 and January 2019. Items were rated individually for content relevance and simplicity on two separate 4-point Likert scales, containing the following options: 1 (Not relevant/simple), 2 (Somewhat relevant/simple), 3 (Quite relevant/simple) and 4 (Highly relevant/simple) [24]. The ratings were dichotomized into agreed (scores 3 and 4) and not agreed (scores 1 and 2). An item level content validity index (I-CVI)

**Table 1. Characteristics of raters in the content validity assessment (Step 2).**

| Rater | Profession | Characteristics |
|-------|-----------|-----------------|
| 1 | Psychologist | Psychology, study population, AITT |
| 2 | Teacher | Pedagogics, study population |
| 3 | Teacher | Pedagogics, study population |
| 4 | Master of Science in Engineering | Study population, AITT |
| 5 | Physician | Family medicine, study population, AITT |
| 6 | Physician, researcher | Family medicine, study population, AITT |
| 7 | Physiotherapist, researcher | Questionnaire development, study population |
| 8 | Paediatric nurse, researcher | Questionnaire development, study population |
| 9 | Nurse, researcher | (Excluded) |

AITT= Assisted infant toilet training.

was calculated with ≥ 0.78 selected as the level for an excellent I-CVI [24,28]. The raters were invited to comment on each item as well as on the comprehensiveness and length of the entire questionnaire. This feedback provided important input to expert panel discussions and to revisions of the draft questionnaire which followed each assessment round. Questions with low scores for relevance and simplicity were discarded or altered. As the BABITT study also carries a longitudinal aspect, a separate question with tick boxes at the end of the domains required validation for additional ages (3, 6, 12 and 18 months as well as 2 and 2.5 years). For example, "Can you also validate these questions for the following ages: 3 months, 6 months?"

With regard to missing data, raters were prompted to complete the information in the few cases where data were missing and the new information was added to the result. No specific analysis of missing data was performed.

### Step 3– Evaluation of feasibility – Pilot phase

In order to evaluate the feasibility of the questionnaire, a 12-month pilot phase was conducted preceding the start of the larger ongoing BABITT study, between the 1st of April 2019 and the 31st of March 2020 [29]. Comments and unanswered questions were readily available for review on the web-platform and response rate was estimated. During the pilot phase, children and parents were invited to a forum at the Child Health Center, where verbal feedback on the questionnaire was solicited. Additionally, input on study participants' perceived ability to complete the questionnaire, was continuously provided by recruiting nurses.

### Ethical statement

The BABITT study was conducted in accordance with the principles of the Declaration of Helsinki. The Swedish Ethical Review Authority approved the BABITT study in November 2018 (Dnr. 2018/388), with complementary approval in April 2019 (Dnr 2019–01668). Before enrollment of study participants, written informed consent was obtained from both parents or guardians. Data has been handled according to European General Data Protection Regulation Services (GDPR).

## Results

### Main elements and construction (Step 1)

**Bowel.**  Due to its great influence on scientific literature, the criteria outlined by the ROME foundation were chosen for the bowel element of the BABITT study and represented the primary outcomes (infant colic, infant dyschezia and functional constipation). By the time our study was planned, the Rome criteria had been updated to their fourth version [30] and the Rome IV Questionnaire on Pediatric Gastrointestinal Disorders (R4PDQ) was being drawn up. Since R4PDQ was not accessible at the time of the BABITT study construction phase, we used the Rome III questionnaire (QPGS-RIII) as a template for items, adapting them in congruence with the updated Rome IV criteria. Also, some alterations were needed to match the study population and parent proxy report format, as QPGS-RIII did not have validated versions for children under 4 years of age. Based on the review of literature, items on "stool toileting refusal" were also added, as well as an item on flatulence.

**Bladder.**  For similar reasons, we chose the ICCS descriptive framework to underpin the bladder element, as it had a high impact on the body of literature [31]. Utilizing the ICCS recommended terminology, items in the bladder element of the questionnaire contain symptoms categorized according to their relation to the storage and voiding phase of bladder function (e.g., voiding frequency, respectively straining at voiding) as well as "other symptoms" including behaviors, such as holding maneuvers. Additional items on dryness and urinary tract infection were included, as relevant to the BABITT study. The reference point for lower urinary tract symptoms was age >5 years, though it was clearly recognized as being selectively applicable to younger cohorts of children, if they were toilet-trained [17]. A clinically useful tool, published as the Bladder-Bowel questionnaire (BBQ), was found to be applicable in the setting [32]. This validated questionnaire in

Swedish, containing 13 items congruent with definitions of bladder dysfunction by ICCS, was used as a template for items in the bladder domain.

**Assisted infant toilet training.** By the time the BABITT study was planned, there were no validated questionnaires on assisted infant toilet training and this new approach to the bowel and bladder research fields provided a new perspective. In addition to expert panel opinion and review of literature, the basis for items included in the questionnaire was derived from the small-scale qualitative interview study conducted prior to the trial [26,27]. The results showed that the toilet training process could be stressful for parents, especially if complications such as constipation arose. Late initiation or disrupted toilet training seemed to be associated with a more trying experience. The parents in the interview study had some knowledge of assisted infant toilet training but were uncertain about how to proceed. The perceived attitude of, and support from the preschool seemed to have a significant impact on the process. Guidance from the Child Health Center was also solicited. The items on assisted infant toilet training in the questionnaire underwent the same development and validation procedures as described in Fig 1. However, they were not included in the figure.

## Content validity assessment (Step 2)

After the final assessment round of the draft questionnaire, I-CVI ranged from 0.88 to 1.00 in most items in all domains for both relevance and simplicity, which was perceived as "excellent" content validity. Generally, somewhat higher scores on relevance than simplicity and significant improvement between assessment rounds emphasized the positive effect of the revisions. The BABITT questionnaire was finalized by a concluding expert panel discussion where the balance between data quality and the respondent burden was carefully considered. Tables 2 and 3 show the I-CVI for two assessment rounds on bowel and bladder items and Table 4 shows items on assisted infant toilet training.

After the second assessment round, items on cow's milk allergy and date for seeking care (bo6, bo13, a14), both scored persistently below cut-off (0.78) and were discarded. However, three items (bo23, bo38 and bl6) were not omitted despite scoring below cut-off. Two of the items (bo23, bo38) pertaining to the same Rome criteria for constipation at age 9 months and 4 years, inquired if the child's stools were unusually large. They received I-CVI scores of 0.75 and 0.63 respectively, with comments such as: *"What is an unusually large diameter? Pictures? Measurements? "*and*"Hard to interpret. 'Unusual' "*. Item (bl6) "straining at voiding", pertaining to ICCS definitions, received 0.75 after the last assessment round. *"What if the child is asked to pee before going out etc.? All of the pee or just at the initiation?"*; *"Never happened with my children."* Likewise, item (bl5) "Hesitancy at start of voiding," received comments of a similar nature but scored above cut-off. The message from the comments was interpreted to mean that it was the very nature of the question that was perceived as awkward, which could not be altered without losing the core information. Taking into account the importance of congruence with diagnostic criteria, the items bo23, bo38 and bl6 remained despite scoring below cut-off.

The item on the definition of infant colic (bo2) was adjusted after the second assessment round by altering answering options. A separate follow-up question was added to address the aspect of time for unexplained crying or fussing. It appeared only when parents reported problems of this nature. Thus, this item was automatically omitted if the parent responded that there was no unexplained crying or fussing.

A ubiquitous comment on answering options was, *"What does **once in a while** an **sometimes** mean? Specify!"* These answering options were altered by adding a percentage as inspired by the construction of other items of the Rome III questionnaire (QPGS-RIII), as follows: Never (0% of the time), Once in a while (25% of the time), Sometimes (50% of the time), Most of the time (75% of the time), Always (100% of the time).

The first version of the draft questionnaire contained 73 items and the second, 74 items. The overall number of items being "too much", was a common remark by raters. To further reduce respondent burden, 20 items were discarded, in the concluding expert panel revision (bo5–6, bo9, bo11–14, bo18, bo26–29, bo33) (bl11, bl19) (a3–4, a13–14, a18) despite scoring above cut-off. Hence, the final version of the BABITT questionnaire contained 55 items, as shown in Table 5. After approval by the expert panel, it was implemented online.

**Table 2. Assessment of bowel items.**

| | Item | Assessment 1 (n= 9) | | | | Assessment 2 (n=8) | | | | Final revision by the expert panel |
|---|---|---|---|---|---|---|---|---|---|---|
| | | Relevance | | Simplicity | | Relevance | | Simplicity | | |
| | | Grade | I-CVI | Grade | I-CVI | Grade | I-CVI | Grade | I-CVI | |
| **BOWEL** | | | | | | | | | | |
| **2 months** | | | | | | | | | | |
| *Infant colic* | | | | | | | | | | |
| Parental perception of colic | bo1 | 3-4 | 1.00 | 3-4 | 1.00 | 4 | 1.00 | 1-4 | 0.86 | B |
| Minutes of crying* | bo2 | 3-4 | 1.00 | 2-4 | 0.67 | 3-4 | 1.00 | 3-4 | 1.00 | AB |
| Days of crying* | bo3 | 3-4 | 1.00 | 3-4 | 1.00 | 3-4 | 1.00 | 3-4 | 1.00 | AB |
| Weeks of crying | bo4 | 3-4 | 1.00 | 2-4 | 0.89 | 3-4 | 1.00 | 3-4 | 1.00 | A |
| Age of onset | bo5 | | | | | 4 | 1.00 | 4 | 1.00 | X |
| Cow's milk allergy | bo6 | | | | | 3-4 | 1.00 | 1-4 | 0.75 | X |
| *Infant dyschezia* | | | | | | | | | | |
| Definition infant dyschezia* | bo7 | 4 | 1.00 | 3-4 | 1.00 | 4 | 1.00 | 2-4 | 0.88 | A |
| Stool consistency* | bo8 | 3-4 | 1.00 | 3-4 | 1.00 | 4 | 1.00 | 2-4 | 0.88 | A |
| Age of onset | bo9 | | | | | 3-4 | 1.00 | 3-4 | 1.00 | X |
| Flatulence | bo10 | 3-4 | 1.00 | 2-4 | 0.88 | 4 | 1.00 | 2-4 | 0.88 | B |
| **9 months** | | | | | | | | | | |
| *Constipation* | | | | | | | | | | |
| Parental concern | bo11 | 3-4 | 1.00 | 2-4 | 0.78 | 4 | 1.00 | 3-4 | 1.00 | X |
| Seeking care | bo12 | | | | | 4 | 1.00 | 2-4 | 0.88 | X |
| Seeking care date | bo13 | | | | | 4 | 1.00 | 2-4 | 0.75 | X |
| Seeking care provider | bo14 | | | | | 1-4 | 0.88 | 2-4 | 0.88 | X |
| Medical treatment frequency | bo15 | 3-4 | 1.00 | 1-4 | 0.89 | 4 | 1.00 | 2-4 | 0.88 | AB |
| Type of medical treatment | bo16 | 3-4 | 1.00 | 2-4 | 0.89 | | | | | |
| Assistance to defecate | bo17 | 3-4 | 1.00 | 2-4 | 0.89 | | | | | |
| Measures to relieve constipation | bo18 | 3-4 | 1.00 | 2-4 | 0.89 | 4 | 1.00 | 3-4 | 1.00 | X |
| Stool frequency* | bo19 | 3-4 | 1.00 | 3-4 | 1 | 4 | 1.00 | 2-4 | 0.88 | A |
| Stool frequency weeks | bo20 | 2-4 | 0.89 | 1-4 | 0.89 | | | | | |
| Pain defecating* | bo21 | 3-4 | 1.00 | 3-4 | 1 | 4 | 1.00 | 2-4 | 0.88 | A |
| Hard stools* | bo22 | 3-4 | 1.00 | 2-4 | 0.78 | 4 | 1.00 | 2-4 | 0.88 | A |
| Diameter of stools* | bo23 | 3-4 | 1.00 | 2-4 | 0.67 | 2-4 | 0.88 | 1-4 | 0.75 | A |
| Stool withholding* | bo24 | 3-4 | 1.00 | 1-4 | 0.78 | 4 | 1.00 | 2-4 | 0.88 | A |
| Fecal mass in examination* | bo25 | 3-4 | 1.00 | 2-4 | 0.89 | 4 | 1.00 | 1-4 | 0.88 | A |
| **4 years** | | | | | | | | | | |
| *Constipation* | | | | | | | | | | |
| Parental concern | bo26 | 3-4 | 1.00 | 2-4 | 0.78 | 4 | 1.00 | 2-4 | 0.88 | X |
| Seeking care | bo27 | | | | | 4 | 1.00 | 3-4 | 1.00 | X |
| Seeking care date | bo28 | | | | | 4 | 1.00 | 1-4 | 0.88 | X |
| Seeking care provider | bo29 | | | | | 1-4 | 0.75 | 2-4 | 0.88 | X |
| Medical treatment frequencyΩ | bo30 | 3-4 | 1.00 | 1-4 | 0.89 | 4 | 1.00 | 2-4 | 0.88 | AB |
| Type of medical treatment | bo31 | 3-4 | 1.00 | 2-4 | 0.89 | | | | | |
| Assistance to defecate | bo32 | 3-4 | 1.00 | 2-4 | 0.89 | | | | | |
| Measures to relieve constipation | bo33 | 3-4 | 1.00 | 2-4 | 0.89 | 4 | 1.00 | 4 | 1.00 | X |
| Stool frequency*Ω | bo34 | 3-4 | 1.00 | 3-4 | 1.00 | 4 | 1.00 | 2-4 | 0.88 | A |
| Stool frequency weeks | bo35 | 2-4 | 0.89 | 1-4 | 0.89 | | | | | |

*(Continued)*

**Table 2.** (Continued)

| | Item | Assessment 1 (n= 9) | | | | Assessment 2 (n=8) | | | | Final revision by the expert panel |
| | | Relevance | | Simplicity | | Relevance | | Simplicity | | |
| | | Grade | I-CVI | Grade | I-CVI | Grade | I-CVI | Grade | I-CVI | |
| Pain defecating* | bo36 | 3-4 | 1.00 | 3-4 | 1.00 | 4 | 1.00 | 2-4 | 0.88 | AB |
| Hard stools*Ω | bo37 | 3-4 | 1.00 | 2-4 | 0.78 | 4 | 1.00 | 2-4 | 0.88 | AB |
| Diameter of stools* | bo38 | 3-4 | 1.00 | 2-4 | 0.67 | 4 | 1.00 | 1-4 | 0.62 | A |
| Stool withholding* | bo39 | 3-4 | 1.00 | 1-4 | 0.78 | 4 | 1.00 | 2-4 | 0.88 | A |
| Fecal mass in examination * | bo40 | 3-4 | 1.00 | 2-4 | 0.89 | 4 | 1.00 | 2-4 | 0.88 | A |
| Clogged toilet* | bo41 | 1-4 | 0.67 | 1-4 | 0.78 | 2-4 | 0.88 | 2-4 | 0.88 | A |
| Procedure when defecating | bo42 | 1-4 | 0.89 | 1-4 | 0.89 | 4 | 1.00 | 2-4 | 0.88 | A |
| Stool toileting refusal | bo43 | 2-4 | 0.89 | 3-4 | 1.00 | 4 | 1.00 | 3-4 | 1.00 | A |
| Stool toileting refusal age | bo44 | 1-4 | 0.78 | 1-4 | 0.89 | 4 | 1.00 | 3-4 | 1.00 | A |

Ratings (grade) and item-level content validity index (I-CVI) for bowel items across the two rounds of assessment and final revision by the expert panel (Step 2).

* =ROME-IV criteria, Ω = Bladder Bowel Questionnaire.

A= Rephrasing of answer options, B= Rephrasing of item, X= Item removed.

## The final BABITT questionnaire

Following the development process, the final BABITT questionnaire consisted of three domains, *Bowel, Bladder* and *Assisted infant toilet training.* It was designed to evaluate bowel and bladder function as well as longitudinally describe the parents' experience of practising assisted infant toilet training.

The *Bowel* domain contains items including diagnostic criteria (Rome IV) for functional gastrointestinal disorders (infant colic, infant dyschezia and/or functional constipation), which are the primary outcomes, evaluated up to 9 months of age in the BABITT study, [33]. The prevalence of infant colic is assessed at ages 2 and 3 months, infant dyschezia at ages 2, 3 and 6 months and functional constipation at ages 2, 3, 6 and 9 months, as well as at 4 years of age. The *Bowel* domain also includes questions on stool toileting refusal and flatulence.

The *Bladder* domain contains items regarding symptoms from the bladder congruent with ICCS definitions [17] as well as descriptive questions on dryness and urinary tract infection up to 4 years of age.

The *Assisted infant toilet training* domain contains items describing the process and parental experiences of toilet training of children up to 2.5 years of age.

English translations of the BABITT questionnaires at 9 months, 2.5 years and 4 years have been supplemented to this paper (S2–S4 File), adding an indication of the content and structure that may aid the readership. As the translation from Swedish into English has not yet been linguistically and culturally validated, the BABITT questionnaire in English should not yet be used for research purposes.

## Feasibility (Step 3)

Response rate at the 2-month questionnaire was 95% as 21 out of 22 families completed it within the given timespan. In the subsequent age groups (3, 6, 9 and 12 months), the response rate was 100%, as all questionnaires were completed during the first year of the study (the two dropouts described below were excluded). This excellent response rate was obtained by automatic reminders dispatched by the web-platform, as well as persistent study coordinator efforts to support and prompt respondents to complete the questionnaires. In the completed questionnaires, no items were left unanswered or omitted by respondents.

**Table 3. Assessment of bladder items.**

| | Item | Assessment 1 (n= 9) | | | | Assessment 2 (n=8) | | | | Final revision by the expert panel |
|---|---|---|---|---|---|---|---|---|---|---|
| | | Relevance | | Simplicity | | Relevance | | Simplicity | | |
| | | Grade | I-CVI | Grade | I-CVI | Grade | I-CVI | Grade | I-CVI | |
| **BLADDER** | | | | | | | | | | |
| **4 years** | | | | | | | | | | |
| *Urinary tract* | | | | | | | | | | |
| Voiding frequency †Ω | bl1 | 3-4 | 1.00 | 2-4 | 0.89 | 4 | 1.00 | 3-4 | 1.00 | A |
| Urgency†Ω | bl2 | 2-4 | 0.89 | 3-4 | 1.00 | 4 | 1.00 | 3-4 | 1.00 | A |
| Postponed voidingΩ | bl3 | 3-4 | 1.00 | 3-4 | 1.00 | 4 | 1.00 | 3-4 | 1.00 | AB |
| Urgency holding maneuver†Ω | bl4 | 2-4 | 0.89 | 2-4 | 0.78 | 4 | 1.00 | 3-4 | 1.00 | AB |
| Hesitancy at start of voiding†Ω | bl5 | 3-4 | 1.00 | 1-4 | 0.78 | | | | | Back after final revision |
| Straining at voiding†Ω | bl6 | 2-4 | 0.89 | 3-4 | 1.00 | 3-4 | 1.00 | 3-4 | 0.75 | A |
| Intermittency at voiding†Ω | bl7 | 3-4 | 1.00 | 3-4 | 1.00 | 3-4 | 1.00 | 2-4 | 0.88 | A |
| Urinary tract infection | bl8 | 4 | 1.00 | 3-4 | 1.00 | 3-4 | 1.00 | 4 | 1.00 | A |
| *Dryness* | | | | | | | | | | |
| Diaper | bl9 | 3-4 | 1.00 | 1-4 | 0.89 | 4 | 1.00 | 3-4 | 1.00 | |
| Diaper daytime | bl10 | 3-4 | 1.00 | 1-4 | 0.89 | | | | | |
| Diaper night time | bl11 | 3-4 | 1.00 | 1-4 | 0.89 | 4 | 1.00 | 2-4 | 0.88 | X |
| Frequency urinary leakage†Ω | bl12 | 3-4 | 1.00 | 1-4 | 0.89 | 4 | 1.00 | 2-4 | 0.88 | AB |
| Extent leakage | bl13 | 3-4 | 1.00 | 1-4 | 0.89 | 4 | 1.00 | 2-4 | 0.88 | A |
| Amount urinary leakageΩ | bl14 | 3-4 | 1.00 | 3-4 | 1.00 | 4 | 1.00 | 4 | 1.00 | A |
| Stool leakage* | bl15 | 3-4 | 1.00 | 1-4 | 0.89 | 4 | 1.00 | 4 | 1.00 | AB |
| Extent stool leakage | bl16 | 3-4 | 1.00 | 3-4 | 1.00 | | | | | |
| Wet morning | bl17 | 3-4 | 1.00 | 3-4 | 1.00 | 4 | 1.00 | 4 | 1.00 | A |
| Night time urination | bl18 | 3-4 | 1.00 | 3-4 | 1.00 | 4 | 1.00 | 3-4 | 1.00 | |
| Urination/night | bl19 | 3-4 | 1.00 | 3-4 | 1.00 | 4 | 1.00 | 3-4 | 1.00 | X |
| Co-sleeping | bl20 | 3-4 | 1.00 | 1-4 | 0.89 | | | | | |
| Age for night dryness | bl21 | 1-4 | 0.89 | 2-4 | 0.89 | 4 | 1.00 | 3-4 | 1.00 | |
| Morning urination | bl22 | 3-4 | 1.00 | 3-4 | 1.00 | 3-4 | 1.00 | 3-4 | 1.00 | A |

Ratings (grade) and item-level content validity index (I-CVI) for bladder items across the two rounds of assessment and final revision by the expert panel (Step 2).

† = ICCS definitions, Ω = Bladder Bowel Questionnaire, * = ROME-IV criteria.

A = Rephrasing of answer options, B = Rephrasing of item, X = Item removed.

There were two dropouts during the 12-month pilot phase, one at 3-month and one at 9-months. Respondent burden to answer the questionnaire was not reported as the main reason for dropping out. In one case, the complexity of the intervention itself was the reason for dropping out and in the other case there were personal reasons.

Data collection and management of the web-questionnaire is comprehensively described in the protocol article [23]. The pilot phase suggested that the questionnaire needed approximately 10 minutes to complete. Many study participants stated that they preferred using their cell phone to access the questionnaire. Unfortunately, the interface was not perceived as user-friendly when using a cell phone, as it was designed for computers. As a result of this finding, the operator adapted the web-survey format for cell phones creating improved conditions for completion of the questionnaire.

A pertinent feature of the web-questionnaire design was automatically omitting redundant questions (i.e., irrelevant to the child's age or due to previous answers), which significantly reduced the respondent burden.

Finally, replying to the questionnaire was not reported to be pivotal for continued participation in the study. In summary, the BABITT questionnaire emerged as valid and feasible in the clinical study setting.

**Table 4. Assessment of assisted infant toilet training items.**

| | Item | Assessment 1 (n=9) | | | | Assessment 2 (n=8) | | | | Final revision by the expert panel |
| | | Relevance | | Simplicity | | Relevance | | Simplicity | | |
| | | Grade | I-CVI | Grade | I-CVI | Grade | I-CVI | Grade | I-CVI | |
| **2,5 years** | | | | | | | | | | |
| *Assisted infant toilet training* | | | | | | | | | | |
| Potting days | a1 | 3-4 | 1.00 | 3-4 | 1.00 | 4 | 1.00 | 2-4 | 0.88 | AB |
| Potting/day | a2 | 3-4 | 1.00 | 2-4 | 0.78 | 4 | 1.00 | 3-4 | 1.00 | A |
| Time of day | a3 | | | | | 3-4 | 1.00 | 3-4 | 1.00 | X |
| Break | a4 | 3-4 | 1.00 | 3-4 | 1.00 | 3-4 | 1.00 | 3-4 | 1.00 | X |
| Pattern stools | a5 | 3-4 | 1.00 | 1-4 | 0.67 | 4 | 1.00 | 1-4 | 0.88 | A |
| Pattern stools description | a6 | 3-4 | 1.00 | 2-4 | 0.78 | | | | | |
| Pattern micturition | a7 | 3-4 | 1.00 | 2-4 | 0.67 | 4 | 1.00 | 2-4 | 0.88 | A |
| Signal stools | a8 | 2-4 | 0.89 | 1-4 | 0.89 | 4 | 1.00 | 2-4 | 0.88 | A |
| Signal micturition | a9 | 2-4 | 0.89 | 2-4 | 0.89 | 4 | 1.00 | 2-4 | 0.88 | A |
| Signal description | a10 | 3-4 | 1.00 | 3-4 | 1.00 | | | | | |
| Holding stools | a11 | 3-4 | 1.00 | 2-4 | 0.67 | 4 | 1.00 | 2-4 | 0.88 | A |
| Holding urine | a12 | | | | | 4 | 1.00 | 2-4 | 0.88 | A |
| Dry stools | a13 | 2-4 | 0.89 | 1-4 | 0.78 | 4 | 1.00 | 2-4 | 0.88 | X |
| Dry urine | a14 | | | | | 4 | 1.00 | 2-4 | 0.63 | X |
| Parental engagement | a15 | 3-4 | 1.00 | 1-4 | 0.78 | 3-4 | 1.00 | 4 | 1.00 | AB |
| Initiative who | a16 | 3-4 | 1.00 | 1-4 | 0.89 | | | | | |
| Initiative child | a17 | | | | | 4 | 1.00 | 3-4 | 1.00 | |
| Initiative adult | a18 | | | | | 4 | 1.00 | 1-4 | 0.88 | X |
| Adult control | a19 | 1-4 | 0.89 | 1-4 | 0.78 | 4 | 1.00 | 3-4 | 1.00 | |
| Experience | a20 | 3-4 | 1.00 | 1-4 | 0.78 | 4 | 1.00 | 3-4 | 1.00 | B |
| Reflections | a21 | 3-4 | 1.00 | 1-4 | 0.78 | 4 | 1.00 | 3-4 | 1.00 | B |

Ratings (grade) and item-level content validity index (I-CVI) for assisted infant toilet training items across the two rounds of assessment and final revision by the expert panel (Step 2).

A= Rephrasing of answer options, B= Rephrasing of item, X= Item removed.

**Table 5. Reduction of items in assessments rounds.**

| | Assessment 1 *Raters n=9* | Assessment 2 *Raters n=8* | Expert panel revision |
| --- | --- | --- | --- |
| Bowel | 35 items | 38 items | 25 items |
| Bladder | 22 items | 18 items | 17 items |
| Assisted infant toilet training | 16 items | 18 items | 13 items |
| **Total** | **73 items** | **74 items** | **55 items** |

Number of items in assessment rounds and after final revisions of the BABITT questionnaire during the validation process.

## Discussion

Combining the research fields of bowel and bladder with assisted infant toilet training, the BABITT questionnaire, congruent with Rome IV criteria and ICCS definitions, was deemed valid according to content validity index, a well-established method for developing of questionnaires [25]. Emerging as feasible in the study context, the Swedish BABITT

questionnaire can now be used to evaluate and describe bowel and bladder function in children as well as assisted infant toilet training up to 4 years of age.

Regarding methodological considerations, merging the four response options into two categories in the content validity assessment (Step 2), has been well established [24,28]. Selecting the nine raters, efforts were made to balance the group composition regarding experience and knowledge of the field as well as ensuring an adequate number of participants [25,28]. A rater displaying significant incongruity with other raters, i.e., by failing to use all the scaling options, could motivate additional clarification of the conditions of the task, as suggested by Grant [25]. However, as the time limits did not permit additional interviewing and training, one rater was excluded from the second assessment round in our study. Hence, the calculation of scores needed slight adjustment for the reduced number of raters (n=8). This loss of input was compensated for, by prolific free-text comments from the remaining group. As the information from two assessment rounds was considered ample to support final revisions, not carrying out a third assessment round was a pragmatic way of keeping within the given timeframe of the study.

Concerning the CVI, low scores for simplicity (as for bl6) might reflect the fact that symptoms of functional disorder are difficult to capture in survey questions in general. However, several items pertaining to Rome IV criteria received remarkably low scores for simplicity, in line with findings suggesting low levels of interrater reliability [34]. This data is consistent with the debate questioning how well the Rome criteria captures pathology in both clinical and non-clinical settings [34–36]. Though iteratively overhauled, the Rome criteria have been considered as cumbersome and even though well defined, as also leaving a lot of room for interpretation [6]. In addition, it is worth noting that the questionnaires provided by the ROME foundation are not freely available. This may pose an obstacle for researchers, due to the time-sensitive nature of scientific trials. Likewise, the gap between updates of the Rome criteria and subsequent production of validated questionnaires also implies challenges. In the present study, for example, we started by using the Rome III questionnaire as a template for bowel items, but this required significant adaptations as the updated Rome IV version was published, during the course of our study.

Due to their status in scientific literature, the Rome criteria and ICCS framework were obvious choices to underpin the bowel and bladder domains. Based on these well-established definitions, the BABITT questionnaire was developed to assess complex conditions (functional bowel and bladder disorders) in relation to the multifaceted parent-child interplay in the toilet training process. The wide range of targeted symptoms and behaviors, as well as the intricacy of the intervention and its interaction with the setting, are the hallmarks of a complex intervention [7]. Conducting qualitative research prior to such an intervention trial provided invaluable input that fuelled the construction phase and generated hypothesis. One could argue that the scope of the study was too wide. On the other hand, complex conditions like bowel and bladder function in relation to a phenomenon as assisted infant toilet training do not allow for too narrow a perspective to be comprehensively described. In addition, the benefit of comparability of results in two adjacent fields of research (bowel and bladder function) in relation to assisted infant toilet training could be considered as enhancing the study's scientific value. Though challenging, engaging in studies which reflect the composite reality is crucial if we wish to achieve clinically useful, evidence-based knowledge. As stated by Skivington, researchers should answer the questions that are most useful to patients and decision makers rather than those that can be answered with greater certainty. "A trade-off always exists between precise unbiased answers to narrow questions and more uncertain answers to broader, more complex questions" [7].

## Strengths and limitations

Ascertaining content validity and feasibility of a questionnaire prior to an intervention trial is highly recommended [29] and is a clear strength of the present study, as well as the ongoing BABITT study [23]. Despite requiring precious additional time and labour, the enhanced validity of the questionnaire was well worth the effort. Furthermore, in accordance with the updated framework of the Medical Research Council for development and evaluation of complex interventions [7],

different perspectives were represented as clinicians, researchers, experts and parents, were engaged throughout the development process, which was a strength giving legacy to the results of our studies.

A literature review was conducted as recommended, to ensure relevance of items [24] and to corroborate the discussions within the expert panel. Admittedly, it could be considered a limitation that the literature review was not performed as a systematic review. However, repeated searches and iterative discussions within the expert panel, which was made up of active researchers in the field of bower and bladder disorders in children, the review of literature was considered sufficiently comprehensive to capture the essentials.

Preceding the start of the BABITT study, no full-scale qualitative study was conducted, which could be considered a limitation. Nevertheless, the results from the small-scale focus group interview study [26,27] provided valuable information during the process of constructing the BABITT questionnaire.

We chose to focus on describing the development process of the bowel and bladder domains in the questionnaire (and not the assisted infant toilet training domain), as they contain the primary outcomes of the BABITT study. This could admittedly be considered as a limitation of the study. However, an outline of the assisted infant toilet training domain and background items was considered sufficient and a sensible choice to improve intelligibility of this paper. Furthermore, to attain an in-depth understanding, future results from the ongoing BABITT study will be analyzed and reported using other (mixed) methods in order to describe the parents' experiences of practicing assisted infant toilet training.

## Conclusion

A web-based questionnaire was developed to evaluate parent-reported bladder and bowel function in children who were introduced to assisted infant toilet training. We conclude that the proposed domains and outline of the BABITT questionnaire emerged as valid in its content by two assessment rounds in the validation process. The CVI method supported the fact that the content of the questionnaire was perceived as relevant, simple and sufficiently comprehensive. Feasibility of the final questionnaire in the clinical study setting was ascertained during the pilot phase. In summary, findings implicate that the questionnaire brings a solid base meeting the scope of the BABITT study.

## Supporting information

**S1 Fig. The development process of the BABITT questionnaire.** The three steps of the development process are graphically visualized. The construction process and main elements in Step 1. The CVI process in two assessment rounds and a final revision in Step 2. The evaluation of feasibility of the final questionnaire in the pilot phase in Step 3.
(TIF)

**S2 File. The BABITT questionnaire 9 months.** English translations of the BABITT questionnaires at 9 months, adding an indication of the content and structure that may aid the readership. As the translation from Swedish into English has not yet been linguistically and culturally validated, it should not be used for research purposes.
(DOCX)

**S3 File. The BABITT questionnaire 2.5 years.** English translations of the BABITT questionnaires at 2.5 years, adding an indication of the content and structure that may aid the readership. As the translation from Swedish into English has not yet been linguistically and culturally validated, it should not be used for research purposes.
(DOCX)

**S4 File. The BABITT questionnaire 4 years.** English translations of the BABITT questionnaires at 4 years, adding an indication of the content and structure that may aid the readership. As the translation from Swedish into English has not yet been linguistically and culturally validated, it should not be used for research purposes.
(DOCX)

# Acknowledgments

The authors would like to thank the families that participated in the pilot phase preceding the start of the larger ongoing BABITT study, the raters participating in the content validity process as well as the study coordinator Eva Eriksson for persistent efforts in supporting data collection.

# Author contributions

**Conceptualization:** Anna Leijon, Terese Nilsson, Linda Vixner, Barbro H. Skogman.

**Data curation:** Anna Leijon.

**Formal analysis:** Anna Leijon, Linda Vixner.

**Funding acquisition:** Anna Leijon, Terese Nilsson, Barbro H. Skogman.

**Methodology:** Anna Leijon, Linda Vixner.

**Project administration:** Anna Leijon.

**Validation:** Anna Leijon, Terese Nilsson, Linda Vixner.

**Visualization:** Anna Leijon.

**Writing – original draft:** Anna Leijon, Barbro H. Skogman.

**Writing – review & editing:** Anna Leijon, Terese Nilsson, Ulla Sillén, Anna-Lena Hellström, Linda Vixner, Barbro H. Skogman.

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
