## [Decision Letter · Decision Letter 0]

30 Dec 2024

PONE-D-24-37861The BABITT questionnaire for evaluation of bowel and bladder function in children who are introduced to assisted infant toilet training - content validity and feasibilityPLOS ONE

Dear Dr. Leijon,

Thank you for submitting your manuscript to PLOS ONE. After careful consideration, we feel that it has merit but does not fully meet PLOS ONE’s publication criteria as it currently stands. Therefore, we invite you to submit a revised version of the manuscript that addresses the points raised during the review process.

We look forward to receiving your revised manuscript.

Kind regards,

Mohd Ismail Ibrahim, MCom.Med

Academic Editor

PLOS ONE

Journal requirements: When submitting your revision, we need you to address these additional requirements. 1. Please ensure that your manuscript meets PLOS ONE's style requirements, including those for file naming. The PLOS ONE style templates can be found at https://journals.plos.org/plosone/s/file?id=wjVg/PLOSOne_formatting_sample_main_body.pdf and https://journals.plos.org/plosone/s/file?id=ba62/PLOSOne_formatting_sample_title_authors_affiliations.pdf. 2. Please amend either the title on the online submission form (via Edit Submission) or the title in the manuscript so that they are identical. 3. We note that the grant information you provided in the ‘Funding Information’ and ‘Financial Disclosure’ sections do not match.  When you resubmit, please ensure that you provide the correct grant numbers for the awards you received for your study in the ‘Funding Information’ section. 4. We note that you have indicated that there are restrictions to data sharing for this study. For studies involving human research participant data or other sensitive data, we encourage authors to share de-identified or anonymized data. However, when data cannot be publicly shared for ethical reasons, we allow authors to make their data sets available upon request. For information on unacceptable data access restrictions, please see http://journals.plos.org/plosone/s/data-availability#loc-unacceptable-data-access-restrictions.  Before we proceed with your manuscript, please address the following prompts: a) If there are ethical or legal restrictions on sharing a de-identified data set, please explain them in detail (e.g., data contain potentially identifying or sensitive patient information, data are owned by a third-party organization, etc.) and who has imposed them (e.g., a Research Ethics Committee or Institutional Review Board, etc.). Please also provide contact information for a data access committee, ethics committee, or other institutional body to which data requests may be sent. b) If there are no restrictions, please upload the minimal anonymized data set necessary to replicate your study findings to a stable, public repository and provide us with the relevant URLs, DOIs, or accession numbers. Please see http://www.bmj.com/content/340/bmj.c181.long for guidelines on how to de-identify and prepare clinical data for publication. For a list of recommended repositories, please see https://journals.plos.org/plosone/s/recommended-repositories. You also have the option of uploading the data as Supporting Information files, but we would recommend depositing data directly to a data repository if possible. Please update your Data Availability statement in the submission form accordingly. 5. In the online submission form, you indicated that [The data thet support the findings of this study are avaliable from Region Dalarna upon reasonable request, provided that the data can be made avaliable in accordance with applicable data protection and privacy regulations.]. All PLOS journals now require all data underlying the findings described in their manuscript to be freely available to other researchers, either 1. In a public repository, 2. Within the manuscript itself, or 3. Uploaded as supplementary information.This policy applies to all data except where public deposition would breach compliance with the protocol approved by your research ethics board. If your data cannot be made publicly available for ethical or legal reasons (e.g., public availability would compromise patient privacy), please explain your reasons on resubmission and your exemption request will be escalated for approval.  6. lease include your full ethics statement in the ‘Methods’ section of your manuscript file. In your statement, please include the full name of the IRB or ethics committee who approved or waived your study, as well as whether or not you obtained informed written or verbal consent. If consent was waived for your study, please include this information in your statement as well.  7. Please upload a copy of S1 Fig 1 to which you refer in your text on page 26 Please amend the file type to 'Supporting Information'. If the Supplementary file is no longer to be included as part of the submission please remove all reference to it within the text.

Reviewers' comments:

Reviewer's Responses to Questions

**Comments to the Author**

1. Is the manuscript technically sound, and do the data support the conclusions?

Reviewer #1: Partly

Reviewer #2: Yes

2. Has the statistical analysis been performed appropriately and rigorously? 

Reviewer #1: Yes

Reviewer #2: N/A

3. Have the authors made all data underlying the findings in their manuscript fully available?

Reviewer #1: No

Reviewer #2: No

4. Is the manuscript presented in an intelligible fashion and written in standard English?

Reviewer #1: Yes

Reviewer #2: Yes

5. Review Comments to the Author

Reviewer #1: 1. Small Sample Size

The study has a very small sample size, which introduces significant limitations. Numerous confounding factors, particularly dietary intake, can significantly impact gut and bladder function. As the study aims to evaluate the effectiveness of toilet training, the questionnaire design should thoroughly account for these confounding factors. These factors should be systematically categorized into distinct groups, with comparisons made only between groups under comparable conditions.

2. Issues with Questionnaire Design

2.1 Single Questionnaire Approach

The study employs multiple questionnaires, which complicates data collection and analysis. A single, unified questionnaire would be a more effective design. For questions that are not relevant to certain age groups, respondents could be instructed to skip them. This approach ensures consistency and simplifies data management as well as future modifications.

For instance, dietary questions for infants primarily consuming milk (under 1 year) differ significantly from those for older children who no longer rely on milk. These dietary-related questions can be grouped into a "Dietary" section, comprising universal questions and age-specific questions. Respondents could skip non-applicable questions based on their child’s age, while an age filter applied during analysis would ensure complete data for each age group.

Similarly, questions related to gut and bladder disorder diagnoses could be organized into a dedicated "Diagnosis" section. Like the dietary questions, respondents could skip questions that do not apply to their child’s age. This clear categorization of sections enhances the structure of the questionnaire, making it more streamlined and facilitating analysis.

This unified approach is particularly well-suited for long-term cohort studies, ensuring data consistency over time while accommodating a wide range of participants.

2.2 Inclusion of Additional Confounding Factors

The current questionnaire omits several critical confounding factors that could influence the study outcomes. A few examples include:

Dietary differences among infants: For infants primarily consuming milk, is there a distinction between those fed breast milk versus formula?

Variations in dietary habits among older children: For 4-year-old children, dietary habits and family environments can vary significantly, even within the same household. Taste preferences can influence gut microbiota composition, directly affecting gut function. Foods or nutrients (e.g., probiotics, yogurt, mushrooms, leafy greens) that alleviate defecation disorders and functional constipation were notably absent from the questionnaire.

Bladder-related factors: For instance, does drinking water before bedtime increase the likelihood of bedwetting?

3. Survey Bias

The study used an "internet-based questionnaire," but the manuscript does not adequately describe the methods of distribution and data collection. Furthermore, it lacks details on how randomization or strict matched design was ensured, which raises concerns about potential survey bias.

4. Baseline Data Table

The manuscript lacks a clear baseline table linking the questionnaire questions to the results. A comprehensive baseline table is essential for understanding the relationship between the variables and outcomes.

5. Conclusion Validity

Given the small sample size, the numerous confounding factors, and the limitations in questionnaire design, the study lacks sufficient evidence to substantiate its conclusions. These methodological constraints significantly weaken the validity of the findings presented in the paper.

Reviewer #2: I had the privilege to review a paper titled The BABITT questionnaire for evaluation of bowel and bladder function in children who are introduced to assisted infant toilet training - content validity and feasibility. This is a very interesting paper, however I have a few comments which need to be clarified.

Study design

The study has not stated what design was used. The authors have just described how the questionnaire was structured and the recruitment process. The first thing that should have been mentioned in the first sentence is the study design.

Results

Check table 2, not all results are showing. Consider using the landscape since this is a wide table.

This also applies to tables 3 and 4.

Conclusion

The conclusion is very brief and has not highlighted the findings of the study. This section must tie the findings together, giving an overall conclusion.

Overall, there are so many limitations, including failure to access the papers that are fundamental to this study, lack of qualitative results except focus group discussions. With all these critical limitations what makes this study to be reliable in terms of internal and external validity?

6. PLOS authors have the option to publish the peer review history of their article (what does this mean? ). If published, this will include your full peer review and any attached files.

**Do you want your identity to be public for this peer review?** For information about this choice, including consent withdrawal, please see our Privacy Policy .

Reviewer #1: No

Reviewer #2: No

---

## [Author Response · Author response to Decision Letter 1]

10 Feb 2025

Firstly, we here give a response to Editors comments:

1. The manuscript meets style requirements for PLOS ONE

2. Titles are checked and congruent

3. Funding Information’ had been corrected but when editing the revision, the box for ‘Financial Disclosure’ was

for some reason not available. The updated information should be:

“The work of conducting the study was funded by grants from the Centre for Clinical Research Dalarna, Region

Dalarna (CKFUU- 961469, CKFUU-933634, CKFUU- 967926) and the Regional Research Council of Uppsala – Örebro (RFR-967829), as well as the Swedish Enuresis Academy, The Samariten Foundation for Paediatric Research and The Mayflower Association. Sponsors and funders have not been involved in the design or conduct of the study nor in analysis of results”

4. a. We have prepared detailed information in Data Availibility Statement with all the requirements

b. N/A

5. We have prepared detailed information in Data Availibility Statement

6. We have now added a section Ethical considerations as the last section under Methods in the manuscript with all required information

7. A copy of S1 Fig 1 will be uploaded as Supportive information

Secondly, we present rebuttals to Reviewers.

Reviewer #1:

1. Small Sample Size

The study has a very small sample size, which introduces significant limitations. Numerous confounding factors, particularly dietary intake, can significantly impact gut and bladder function. As the study aims to evaluate the effectiveness of toilet training, the questionnaire design should thoroughly account for these confounding factors. These factors should be systematically categorized into distinct groups, with comparisons made only between groups under comparable conditions.

Answer:

There seems to be a misconception about sample size in this present study (n=22), and we hereby hope to clarify the issue. The aim of the lager ongoing BABITT study (n=293), as described in the protocol paper, is to evaluate the effectiveness of assisted infant toilet training (Nilsson T et al 2022). The sample size in the larger BABITT study matches the power calculation, allowing a comparison between groups, and all relevant remarks mentioned above will be addressed in future publications. In this present study, we included only 22 patients during the pilot phase (preceding the start of the ongoing BABITT study) in order to evaluate feasibility of the questionnaire and therefor a sample size calculation was not relevant.

Thus, the pilot phase is a separate part preceding the BABITT study, which now has been clarified in Abstract (line 31-32) and in the main manuscript (line 196).

2. Issues with Questionnaire Design

2.1 Single Questionnaire Approach

The study employs multiple questionnaires, which complicates data collection and analysis. A single, unified questionnaire would be a more effective design. For questions that are not relevant to certain age groups, respondents could be instructed to skip them. This approach ensures consistency and simplifies data management as well as future modifications.

For instance, dietary questions for infants primarily consuming milk (under 1 year) differ significantly from those for older children who no longer rely on milk. These dietary-related questions can be grouped into a "Dietary" section, comprising universal questions and age-specific questions. Respondents could skip non-applicable questions based on their child’s age, while an age filter applied during analysis would ensure complete data for each age group.

Similarly, questions related to gut and bladder disorder diagnoses could be organized into a dedicated "Diagnosis" section. Like the dietary questions, respondents could skip questions that do not apply to their child’s age. This clear categorization of sections enhances the structure of the questionnaire, making it more streamlined and facilitating analysis.

This unified approach is particularly well-suited for long-term cohort studies, ensuring data consistency over time while accommodating a wide range of participants.

Answer:

Indeed, this an important and relevant remark from reviewer #1, which is also addressed in the discussion part of the manuscript. A single, unified questionnaire would admittedly have been a more manageable questionnaire. However, we choose a wider perspective in this matter based on the argumentation by Skivington K et al 2021 on the importance of complex interventions. We now further elaborate on this in the Discussion part of the manuscript (line 413-416).

Concerning the reviewers comment about non-applicable questions in the questionnaire, the design of the web-based questionnaire did automatically allow respondents to skip questions irrelevant due to age or the nature of previous answers. As the reviewers comments highlight, this significantly will facilitate analysis of results. This is now more clearly described in the manuscript (line 364-366).

2.2 Inclusion of Additional Confounding Factors

The current questionnaire omits several critical confounding factors that could influence the study outcomes. A few examples include:

Dietary differences among infants: For infants primarily consuming milk, is there a distinction between those fed breast milk versus formula?

Variations in dietary habits among older children: For 4-year-old children, dietary habits and family environments can vary significantly, even within the same household. Taste preferences can influence gut microbiota composition, directly affecting gut function. Foods or nutrients (e.g., probiotics, yogurt, mushrooms, leafy greens) that alleviate defecation disorders and functional constipation were notably absent from the questionnaire.

Bladder-related factors: For instance, does drinking water before bedtime increase the likelihood of bedwetting?

Answer:

Additional items on dietary habits and family environments (for example feeding, parental education and family situation) are indeed relevant to the study and are included in the BABITT questionnaire. These items were also assessed by the CVI method in this present study, however not included in this paper due to legibility (line 168-169).

All confounding factors possibly influencing the outcomes in the larger ongoing BABITT study, as mentioned above in reviewers comments, are interesting and relevant and will be addressed in future publications. However, the RCT design of the BABITT study eliminates influence of confounding factors on the outcome. This present study does not compare outcomes in groups and confounding factors are not relevant. It has now been clarified that the larger ongoing BABITT study is randomized (line 27 and 97).

3. Survey Bias

The study used an "internet-based questionnaire," but the manuscript does not adequately describe the methods of distribution and data collection. Furthermore, it lacks details on how randomization or strict matched design was ensured, which raises concerns about potential survey bias.

Answer:

There seems to be a misconception about study design. This present study is an observational study, which has now been added in Abstract (line 31-32), in the last sentence in Introduction (line 107) and in the first sentence under Method/Study design in the manuscript (line 113-114).

Furthermore, the relationship between in the present study, with the pilot phase preceding the start of the larger ongoing BABITT study, is clarified and we have made adjustments accordingly in Abstract (line 31-32) and in the main manuscript (line 196).

Data collection and management is elaborately described in the protocol article (Nilsson T el al. 2022) and information is now added (line 357-359).

4. Baseline Data Table

The manuscript lacks a clear baseline table linking the questionnaire questions to the results. A comprehensive baseline table is essential for understanding the relationship between the variables and outcomes.

Answer:

This present study is an observational study about the development of a questionnaire. The variables and outcomes of the BABITT study (RCT) will be reported in future papers. For clarity, we made adjustments as mentioned above (line 31-32, 107 and 113-114).

5. Conclusion Validity

Given the small sample size, the numerous confounding factors, and the limitations in questionnaire design, the study lacks sufficient evidence to substantiate its conclusions. These methodological constraints significantly weaken the validity of the findings presented in the paper.

Answer:

Sample size and confunding factors are discussed above, and only relevant for the BABITT study. The relationship between the this observational study and the larger ongoing BABITT study (RCT) is now clarified in the manuscript (line 31-32, 107 and 113-114)

Reviewer #2

I had the privilege to review a paper titled The BABITT questionnaire for evaluation of bowel and bladder function in children who are introduced to assisted infant toilet training - content validity and feasibility. This is a very interesting paper, however I have a few comments which need to be clarified.

Study designThe study has not stated what design was used. The authors have just described how the questionnaire was structured and the recruitment process. The first thing that should have been mentioned in the first sentence is the study design.

Answer:

Thank you for this comment about study design. An observational study design with expert assessments was used to construct, assess content validity and evaluate feasibility of a questionnaire for parent report. . This is now added as first sentence under Method/Study design (line 113-114).

Results

Check table 2, not all results are showing. Consider using the landscape since this is a wide table. This also applies to tables 3 and 4.

Answer:

Table 2,3,4 has now been corrected.

Conclusion

The conclusion is very brief and has not highlighted the findings of the study. This section must tie the findings together, giving an overall conclusion.

Overall, there are so many limitations, including failure to access the papers that are fundamental to this study, lack of qualitative results except focus group discussions. With all these critical limitations what makes this study to be reliable in terms of internal and external validity?

Answer:

Thank you for valuable and wise input. We have now elaborated on the conclusion, tying the findings together, giving an ample content and hopefully providing the reader a more complete overall conclusion (line 454-466).

Addressing your attentive comments on failure to access papers, references to the small-scale qualitative interview studies performed by the researchers (AL and TN) are now included in reference list (line 543-550, reference numbers 26 and 27).

Apart from our focus group interviews, no results from further qualitative studies were available in the field of assistant toilet training at the time, hence could unfortunately not be referred to in our manuscript.

Admittedly, there are limitations in our study being addressed in the Strength and Limitations in the discussion part of the manuscript. Concerning external validity, we would argue that including questions from previously validated questionnaires (the Rome IV Questionnaire on Pediatric Gastrointestinal Disorders (R4PDQ) and Bladder-Bowel questionnaire (BBQ)) vouches for the external validity of the BABITT questionnaire. Choosing these instruments representing gold standard in the research field of bowel and bladder disorders was, in our perspective, the best way of guaranteeing external validity to our BABITT questionnaire.

Concerning internal validity of the final version of the BABITT questionnaire, it is warranted by the CVI process described in this present paper.

---

## [Decision Letter · Decision Letter 1]

21 Feb 2025

The BABITT questionnaire for evaluation of bowel and bladder function in children who are introduced to assisted infant toilet training - content validity and feasibility

PONE-D-24-37861R1

Dear Dr. Leijon,

We’re pleased to inform you that your manuscript has been judged scientifically suitable for publication and will be formally accepted for publication once it meets all outstanding technical requirements.

Kind regards,

Mohd Ismail Ibrahim, MCom.Med

Academic Editor

PLOS ONE

Additional Editor Comments (optional):

Reviewers' comments:

Reviewer's Responses to Questions

**Comments to the Author**

1. If the authors have adequately addressed your comments raised in a previous round of review and you feel that this manuscript is now acceptable for publication, you may indicate that here to bypass the “Comments to the Author” section, enter your conflict of interest statement in the “Confidential to Editor” section, and submit your "Accept" recommendation.

Reviewer #1: All comments have been addressed

2. Is the manuscript technically sound, and do the data support the conclusions?

Reviewer #1: Yes

3. Has the statistical analysis been performed appropriately and rigorously? 

Reviewer #1: Yes

4. Have the authors made all data underlying the findings in their manuscript fully available?

Reviewer #1: Yes

5. Is the manuscript presented in an intelligible fashion and written in standard English?

Reviewer #1: Yes

6. Review Comments to the Author

Reviewer #1: The revised version provides a more precise description of the experimental design and addresses the concerns raised.

PS. The formatting of the PDF looks strange after adjustment.

7. PLOS authors have the option to publish the peer review history of their article (what does this mean? ). If published, this will include your full peer review and any attached files.

**Do you want your identity to be public for this peer review?** For information about this choice, including consent withdrawal, please see our Privacy Policy .

Reviewer #1: No

---

## [Editor Report · Acceptance letter]

PONE-D-24-37861R1

PLOS ONE

Dear Dr. Leijon,

I'm pleased to inform you that your manuscript has been deemed suitable for publication in PLOS ONE. Congratulations! Your manuscript is now being handed over to our production team.

Kind regards,

on behalf of

Dr. Mohd Ismail Ibrahim

Academic Editor

PLOS ONE
